# Clinical Relevance of Collagen Protein Degradation Markers C3M and C4M in the Serum of Breast Cancer Patients Treated with Neoadjuvant Therapy in the GeparQuinto Trial

**DOI:** 10.3390/cancers11081186

**Published:** 2019-08-15

**Authors:** Malgorzata Banys-Paluchowski, Sibylle Loibl, Isabell Witzel, Christoph Mundhenke, Bianca Lederer, Christine Solbach, Thomas Karn, Frederik Marmé, Valentina Nekljudova, Christian Schem, Elmar Stickeler, Nicholas Willumsen, Morten A. Karsdal, Michael Untch, Volkmar Müller

**Affiliations:** 1Department of Gynecology and Obstetrics, Asklepios Klinik Barmbek, 22307 Hamburg, Germany; 2German Breast Group, 63263 Neu-Isenburg, Germany; 3Department of Gynecology, University of Hamburg-Eppendorf, 20251 Hamburg, Germany; 4Department of Gynecology and Obstetrics, University of Kiel, 24105 Kiel, Germany; 5Department of Gynecology and Obstetrics, University of Frankfurt, 60590 Frankfurt am Main, Germany; 6University Hospital Mannheim, Medical Faculty Mannheim of the Heidelberg University, 68167 Mannheim, Germany; 7Mammazentrum Hamburg, 20357 Hamburg, Germany; 8Department of Gynecology and Obstetrics, RWTH Aachen University, 52074 Aachen, Germany; 9Nordic Bioscience, Biomarkers and Research, 2730 Herlev, Denmark; 10Department of Gynecology and Obstetrics, Helios Klinikum Berlin-Buch, 13125 Berlin, Germany

**Keywords:** breast cancer, C3M, C4M, collagen degradation marker, neoadjuvant therapy, trastuzumab, lapatinib

## Abstract

*Background*: Remodeling of extracellular matrix through collagen degradation is a crucial step in the metastatic cascade. The aim of this study was to evaluate the potential clinical relevance of the serum collagen degradation markers (CDM) C3M and C4M during neoadjuvant chemotherapy for breast cancer. *Methods*: Patients from the GeparQuinto phase 3 trial with untreated HER2-positive operable or locally advanced breast cancer were enrolled between 7 November 2007, and 9 July 2010, and randomly assigned to receive neoadjuvant treatment with EC/docetaxel with either trastuzumab or lapatinib. Blood samples were collected at baseline, after four cycles of chemotherapy and at surgery. Cutoff values were determined using validated cutoff finder software (C3M: Low ≤9.00 ng/mL, high >9.00 ng/mL, C4M: Low ≤40.91 ng/mL, high >40.91 ng/mL). Results: 157 patients were included in this analysis. At baseline, 11.7% and 14.8% of patients had high C3M and C4M serum levels, respectively. No correlation was observed between CDM and classical clinical-pathological factors. Patients with high levels of CDM were significantly more likely to achieve a pathological complete response (pCR, defined as ypT0 ypN0) than patients with low levels (C3M: 66.7% vs. 25.7%, *p* = 0.002; C4M: 52.7% vs. 26.6%, *p* = 0.031). Median levels of both markers were lower at the time of surgery than at baseline. In the multivariate analysis including clinical-pathological factors and C3M levels at baseline and changes in C3M levels between baseline and after four cycles of therapy, only C3M levels at baseline (*p* = 0.035, OR 4.469, 95%-CI 1.115–17.919) independently predicted pCR. In a similar model including clinical-pathological factors and C4M, only C4M levels at baseline (*p* = 0.028, OR 6.203, 95%-CI 1.220–31.546) and tumor size (*p* = 0.035, OR 4.900, 95%-CI 1.122–21.393) were independent predictors of pCR. High C3M levels at baseline did not correlate with survival in the entire cohort but were associated with worse disease-free survival (DFS; *p* = 0.029, 5-year DFS 40.0% vs. 74.9%) and overall survival (OS; *p* = 0.020, 5-year OS 60.0% vs. 88.3%) in the subgroup of patients randomized to lapatinib. In the trastuzumab arm, C3M did not correlate with survival. In the entire patient cohort, high levels of C4M at baseline were significantly associated with shorter DFS (*p* = 0.001, 5-year DFS 53.1% vs. 81.6%) but not with OS. When treatment arms were considered separately, the association with DFS was still significant (*p* = 0.014, 5-year DFS 44.4% vs. 77.0% in the lapatinib arm; *p* = 0.023, 5-year DFS 62.5% vs. 86.2% in the trastuzumab arm). *Conclusions*: Collagen degradation markers are associated with response to neoadjuvant therapy and seem to play a role in breast cancer.

## 1. Introduction

The interaction between cancer cells and their microenvironment is considered a crucial factor in the metastatic cascade, allowing tumor cells to proliferate, build new vessels, leave the primary tumor bed and finally enter and persist at secondary homing sites. The non-cellular part of the tumor microenvironment, the extracellular matrix (ECM), consists of a variety of macromolecules, such as collagen and glycoproteins [1]. While the basement membranes of the ECM are formed mostly by type IV collagen, type I and type III collagen are the most abundant proteins of the underlying interstitial matrix. ECM undergoes constant remodeling, mediated mainly by matrix-metalloproteinases (MMP), and in the healthy tissue, matrix degradation is balanced by protein formation. This controlled homeostasis is assumed to be disrupted during cancer development and progression [2].

In the process of MMP-mediated ECM degradation, small fragments of ECM turnover products are generated and released into the bloodstream. Several studies have shown that serum levels of collagen degradation fragments are elevated in cancer patients compared to healthy controls [3,4]. Bager et al. found levels of MMP-degraded collagen type I, III and IV (i.e., C1M, C3M and C4M, respectively) to be 1.5 to 6-fold higher in ovarian and breast cancer patients than in controls [3]. Similarly, elevated levels of serum biomarkers reflecting altered collagen turnover have been reported in colorectal and lung cancer as well as melanoma [5,6]. In pancreatic cancer, a combined panel of serum collagen degradation markers was able to differentiate patients from healthy controls with a high diagnostic power [7].

So far, evidence regarding the prognostic significance of soluble ECM turnover markers is still limited. Recently, Lipton et al. showed that elevated levels of MMP-generated collagen fragments in the serum of metastatic breast cancer patients were associated with shorter time to progression and overall survival [8]. In a Danish register-based study, C1M but not C4M was predictive of survival in postmenopausal women diagnosed with any cancer within 3 years of blood sampling [9]. However, no study has focused on the role of collagen degradation fragments in women with early breast cancer.

The aims of the present study were 1) to assess the clinical relevance of serum C3M and C4M in patients with HER2-positive breast cancer scheduled to receive neoadjuvant therapy and 2) to examine the changes of these markers during systemic treatment.

## 2. Results

### 2.1. Serum Levels of Collagen Degradation Markers

A total of 615 HER2-positive patients were treated in the GeparQuinto trial (Figure 1). Out of these, 157 patients were included in this substudy and provided blood samples (Figure 2). Arms were well balanced with no significant differences between patients treated with trastuzumab and lapatinib regarding age or tumor characteristics (Appendix A
Table A1).

At baseline, levels of collagen degradation markers C3M and C4M were determined in 128 patients (Figure 2). A total of 15 out of 128 patients (12%) had high C3M levels (i.e., >9.004 ng/mL) and 19 out of 128 patients (15%) had high C4M levels (i.e., >40.91 ng/mL). Clinical-pathological data are summarized in Table 1. Compared to serum levels in patients, healthy females had lower levels of both C3M and C4M (C3M: Median 6.4 vs. 3.6 ng/mL; C4M: Median 31.9 vs. 25.3 ng/mL, respectively). Levels of collagen degradation markers in breast cancer patients did not correlate with established clinical-pathological factors such as hormone receptor status or grading (Table 1).

Further measurements of collagen degradation markers were conducted after four cycles of neoadjuvant chemotherapy with epirubicin and cyclophosphamide and at surgery. Median levels of both markers at surgery were lower than at baseline (Table 2). Interestingly, while median levels of C3M were lower after four cycles than before the start of treatment, we observed an initial increase in median C4M levels between baseline and after four cycles of therapy. C3M and C4M levels increased between baseline and after four therapy cycles in 44.4% and 52.9% of patients, respectively. The proportion of patients with an increase in C3M levels between baseline and after four cycles was comparable in the trastuzumab and lapatinib arm (44% and 45%, respectively), while more patients experienced an increase in C4M levels when treated with trastuzumab than lapatinib (58% vs. 48%, respectively).

An increase of at least 20% in serum C3M and C4M levels between baseline and surgery was seen in 23% and 24% of patients, respectively (Table 3), whereas serum levels declined by at least 20% in 37% in case of C3M and 22% in case of C4M. The changes in serum marker levels were similar between treatment arms.

### 2.2. Collagen Degradation Markers and Response to Therapy

Patients with elevated serum levels of C3M and C4M were significantly more likely to achieve pathological complete response (pCR). Out of 15 patients with high C3M levels at baseline, ten (66.7%) achieved pCR, compared to 29 out of 113 (25.7%) patients with low levels (*p* = 0.002, whereas 10 out of 19 (52.6%) of patients with high C4M levels and 29 out of 109 (26.6%) of patients with low C4M levels achieved pCR (*p* = 0.031) (Table 1). In the multivariate analysis including pre-chemotherapy tumor size, nodal status, grading, hormone receptor status, C3M levels at baseline and changes in C3M levels between baseline and after four cycles of therapy, only C3M levels at baseline (*p* = 0.035, OR 4.469, 95%-CI 1.115–17.919) independently predicted pCR. In a multivariate model including pre-chemotherapy tumor size, nodal status, grading, hormone receptor status, C4M levels at baseline and changes in C4M levels between baseline and after four cycles of therapy, only C4M levels at baseline (*p* = 0.028, OR 6.203, 95%-CI 1.220–31.546) and tumor size (*p* = 0.035, OR 4.900, 95%–CI 1.122–21.393) were independent predictors of pCR.

### 2.3. Collagen Degradation Markers and Survival

After a median follow up of 60.6 months, 14 patients died and 34 were diagnosed with a relapse. High C3M levels at baseline did not correlate with survival in the entire cohort (Figure 3, Table 4) but were associated with worse disease-free survival (DFS; *p* = 0.029, 5-year DFS 40.0% vs. 74.9%) and overall survival (OS; *p* = 0.020, 5-year OS 60.0% vs. 88.3%) in the subgroup of patients randomized to lapatinib. In the trastuzumab arm, C3M did not correlate with survival (DFS: *p* = 0.775, 5-year DFS 75.0% vs. 84.2%; OS: *p* = 0.678, 5-year OS 87.5% vs. 91.7%). In the entire patient cohort, high levels of C4M at baseline were significantly associated with shorter DFS but not with OS (Figure 4, Table 4). When treatment arms were considered separately, the association with DFS was still significant (*p* = 0.014 and 5-year DFS 44.4% vs. 77.0% in the lapatinib arm; *p* = 0.023 and 5-year DFS 62.5% vs. 86.2% in the trastuzumab arm).

A decline of at least 20% in serum levels of C3M between baseline and after four cycles of therapy predicted significantly shorter DFS (*p* = 0.004, HR 3.55, 95%-CI 1.51–8.31) and OS (*p* = 0.022, HR 6.49, 95%-CI 1.31–32.23) (Figure 5, Appendix A
Table A2). No association between changes in C4M levels and survival were observed.

## 3. Discussion

To the best of our knowledge, this is the first study to examine collagen degradation markers in the serum of patients undergoing neoadjuvant therapy for breast cancer. At baseline, patients showed higher levels of C3M and C4M than healthy controls. This is in accordance with previous studies. Bager et al. investigated biomarkers reflecting altered MMP-mediated collagen turnover and reported significantly lower levels of both serum markers in healthy women than in breast and ovarian cancer patients [3], suggesting that the tightly controlled homeostasis of the ECM can be severely disturbed by malignant growth. Indeed, alterations of the microenvironment seem to play a crucial role in cancer development and progression. While the major fibrillar collagens have a relatively slow metabolic turnover in histologically normal breast tissue and in benign lesions, increased expression of various types of procollagen mRNAs was reported in the fibroblastic cells of the stroma surrounding breast cancer cells [10]. A comprehensive gene expression portrait of cells composing breast tissue in situ and invasive breast carcinomas showed significant changes in gene expression profile in all cell types during tumor progression, indicating that the microenvironment actively participates in cancer growth and invasion [11].

Interestingly, baseline levels of collagen degradation markers did not correlate with clinical-pathological factors. However, all patients included in the present analysis had HER2-positive disease since the trial was aimed at comparing trastuzumab and lapatinib in the neoadjuvant setting. Whether serum levels of ECM-related biomarkers differ between tumor subtypes, remains therefore unclear. Previously, Bergamaschi et al. showed in a microarray-based study that tumors can be divided into four subgroups based on the expression of the components of extracellular matrix and that these ECM signatures showed low correlation with five intrinsic subtypes [12]. In the group of patients with luminal tumors assessment of ECM signature identified those with a particularly poor prognosis. In addition, it has been hypothesized that HER2-mediated pathways may influence ECM degradation. Data from gastric and breast cancer suggest a cross-talk between various members of ECM and HER2. In a gastric cancer study, HER2 knockdown led to downregulation of the expression of MMP-1, while HER2 overexpression enhanced the transcription of MMP-1 through the activation of a specific promoter [13]. In breast cancer (BC), the addition of MMP-9 to cell lines resulted in a significant rise in HER2 expression, indicating that some MMPs may serve as regulators of HER2 expression on human epithelial cells [14]. Another important finding is the observation that a large proportion of patients with HER2-positive BC develop resistance to trastuzumab and this might be due to the loss of expression of HER2 extracellular domain on tumor cells, caused by shedding/cleavage of HER2 by metalloproteinases [15].

Furthermore, in our study patients with elevated levels of C3M and C4M at baseline were significantly more likely to achieve a pathologically complete response. Previous studies have shown that tumor microenvironment can contribute to response or resistance to chemotherapeutic agents through different mechanisms [16]. For example, dense ECM can limit the blood flow and thus restrict access of the drug to parts of the tumor; further, interactions of tumor cells with ECM proteins may activate pro-survival signaling cascades leading to immediate drug protection or stromal cells may secrete pro-survival factors [17]. These mechanisms may allow tumor cells to survive long enough to acquire genetic changes leading to drug resistance. Jansen et al. examined tumor tissue from 112 patients with ER-positive metastatic BC treated with first-line tamoxifen and showed that six genes associated with the ECM (*TIMP3*, *FN1*, *LOX*, *COL1A1*, *SPARC* and *TNC*) were overexpressed in patients with resistant disease [18]. A translational substudy within the EORTC 10994/BIG 00-01 trial aimed at assessing the relevance of stromal cell expression in biopsy specimens before the start of neoadjuvant therapy with FEC [19]. The expression of stromal metagenes significantly predicted a response to therapy in this trial as well as in an independent cohort of ER-negative tumors from the M. D. Anderson Medical Center included in another neoadjuvant study [20]. However, in contrast to our trial, the above-mentioned studies focused on the ECM-related gene expression in tumor tissue and not on proteins in the blood. To the best of our knowledge, the only other study on the response to therapy and survival in the context of ECM degradations markers in breast cancer was conducted in the metastatic setting. Lipton et al. measured C1M, C3M, C4M and PRO-C3 in the pre-treatment serum of patients from a phase III randomized trial of 2nd-line endocrine therapy (n = 148) and a 1st-line trastuzumab-treated cohort (n = 55) [8]. In the HR-positive cohort, higher C1M and C3M levels were associated with shorter time to progression; all fragments were associated with a shorter OS. In multivariate analysis for OS, higher levels of all fragments were significant for a reduced OS when added separately. In the HER2-positive cohort, higher levels of all fragments were associated with shorter time to progression; higher PRO-C3 was associated with a shorter OS. Indeed, patients with elevated levels of C3M and C4M in our study had numerically shorter DFS and OS, in case of C4M and OS the difference was statistically significant. Interestingly, higher levels of collagen degradation markers predicted both pCR and shorter survival in our study. While surprising at first, this observation suggests that a high turnover of the extracellular matrix is a typical feature of aggressive tumor behavior.

With regard to changes in serum degradation markers during systemic therapy, we show that levels of C3M and C4M decline between the start of treatment and surgery. Interestingly, a strong decline of C3M during the first four cycles of therapy was associated with worse clinical outcome. Possibly, this observation is confounded by the fact that patients with initially high levels of serum markers are more likely to experience a decline. This would further support our hypothesis that elevated levels of degradation markers reflect a more aggressive type of tumor. A small study investigated levels of collagen IV in the serum of 51 breast cancer patients undergoing neoadjuvant therapy and found these to be higher than in healthy controls [21]. Collagen IV levels increased during therapy but were not predictive of response to therapy. However, since we detected collagen degradation markers and not collagen levels, it is unclear how these older data should be interpreted in the light of our findings. Whether serum degradation markers or components of the extracellular matrix might serve as a potential therapeutic target, remains also unclear [22,23].

The weak points of our analysis are the small sample size and the fact that serum samples at three predefined time points were not collected in all 157 potentially available patients. The participation in this substudy was voluntary and had no influence on the participation in the phase III clinical trial; further, there were other translational subprojects available. Secondly, all patients had HER2-positive tumors. The clinical trial GeparQuinto was designed to evaluate differences in the efficacy between trastuzumab and lapatinib. A marker able to identify patients who benefit more from a tyrosine kinase inhibitor like lapatinib would be of clinical relevance. We decided to examine two known collagen degradation markers because these were deemed most promising in the context of predicting therapy response. However, levels of degradation markers in the serum did not predict which HER2-targeted therapy the patients are most likely to benefit from.

## 4. Materials and Methods 

The design of the GeparQuinto phase 3 trial (NCT00567554, EudraCT registration number 2006-005834-19) was described in detail elsewhere [24,25,26]. Briefly, patients with untreated HER2-positive operable or locally advanced breast cancer were enrolled between 7 November 2007, and 9 July 2010 and randomly assigned to receive neoadjuvant treatment with four cycles of EC (epirubicin 90 mg/m^2^, cyclophosphamide 600 mg/m^2^, day 1, every 3 weeks) followed by four cycles of docetaxel (100 mg/m^2^, day 1, every 3 weeks) with either trastuzumab (6 mg/kg intravenously, every 3 weeks, starting with a loading dose of 8 mg/kg) or lapatinib (1250 mg per day) throughout all cycles before surgery. Patients completed 1 year of anti-HER2 treatment with trastuzumab after surgery in both treatment groups.

Blood samples were collected at three different time points: Before initiation of neoadjuvant therapy, after four cycles of EC and prior to surgery (Figure 1). Samples were stored at −20 °C and shipped on dry ice. All patients gave written informed consent to blood collection as part of the correlative science program, before entering the study. Participation in the clinical trial was still possible even if a patient did not agree to participate in the translational research project. The biomarker investigations in the GeparQuinto study were performed after approval by the Institutional Review Board (IRB) (Registration number: 4 December 2007, Ethics commission of the special field “Medicine” at the Goethe University Frankfurt, Germany; EudraCT number: 2006-005834-19; Date of registration: 12 April 2007). Samples from healthy controls included blood samples from 19 females and were part of a previously published study (Appendix A
Table A3) [27].

Trial registration: NCT00567554, EudraCT registration number 2006-005834-19. This study was conducted by the German Breast Group and the AGO-Breast.

### 4.1. Quantitative Analysis of C3M and C4M

Serum markers were measured using a competitive Enzyme-Linked Immune Sorbent Associated (ELISA) assay (Nordic Bioscience, Herlev, Denmark) according to the manufacturer’s specifications. The validation steps for these assays have been described in detail previously [28,29]. Since C3M and C4M are generated by different MMPs, they specifically reflect MMP-mediated degradation of type III and type IV collagen, respectively. Each collagen fragment has a specific protease-generated neo-epitope (cleavage site) present against which the individual immuno-assay is highly specific. The intraassay and interassay variations were <10% and <15%, respectively.

### 4.2. Statistical Analysis

Fisher’s Exact Test was used to examine the relationship between C3M/C4M levels and other parameters, such as clinical and histological factors. Univariate and multivariate logistic regressions were used to determine odds ratios and 95% CI for pCR (defined as ypT0 ypN0) according to the biomarker.

A publicly available Cutoff Finder software (http://molpath.charite.de/cutoff/) was used to convert the continuous variables into dichotomous variables with the outcome variable pCR for all analyses (C3M low ≤9.004 ng/mL, high >9.004 ng/mL, C4M: low ≤40.91 ng/mL, high >40.91 ng/mL) [30]. Using these cutoff values, 11.7% and 14.8% of patients had high C3M and C4M levels at baseline, respectively. Levels of serum degradation markers were measured in a cohort of 19 healthy females with a mean age of 51 years (C3M: Mean 3.9 ng/mL, median 3.6 ng/mL; C4M: Mean 28.2 ng/mL, median 25.3 ng/mL). With regard to changes in serum degradation markers, “no change” was defined as an increase <20% or a decrease <20%.

All time-to-event endpoints were defined as the time (in months) from random assignment to the event; patients without an event were censored at the time of the last contact. Events for DFS were any loco-regional (ipsilateral breast or local/regional lymph nodes) recurrence of disease, any contralateral breast cancer, any distant recurrence of disease, any secondary malignancy or death as a result of any cause, whichever occurred first. OS was defined as the time since random assignment until death as a result of any cause [31]. Kaplan–Meier product limit method was used to estimate the time-to-event curves. The log-rank test was used to evaluate the univariate significance of the clinical, histological and blood-based parameters for the time-to-event endpoints. Cox proportional hazard models were constructed to evaluate the multivariate significance.

All reported p-values are two-sided; values ≤0.05 were considered significant. Statistical analysis was performed using the SPSS program Version 21.0 (SPSS Inc., Chicago, IL, USA). The study was performed and the manuscript prepared according to the REporting recommendations for tumor MARKer prognostic studies (REMARK) criteria on reporting of biomarkers [32].

## 5. Conclusions

In the present study, we show that levels of collagen degradation markers in the serum of BC patients are higher than in healthy controls. Patients with elevated C3M and C4M levels were more likely to achieve pCR; however, high levels of CDM were also associated with worse clinical outcome. In summary, our results suggest that enhanced ECM remodeling reflected by serum degradation markers in early breast cancer is associated with more aggressive and proliferative biology. Potentially, the measurement of collagen degradation markers might help to identify patients in need of additional secondary adjuvant therapeutic strategies.

## Figures and Tables

**Figure 1 cancers-11-01186-f001:**
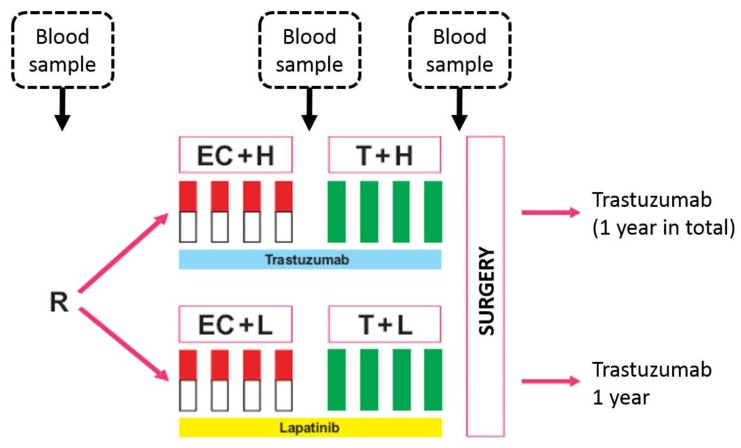
Serum collection in the HER2-positive cohort in the GeparQuinto trial.Abbreviations: EC–epirubicin/cyclophosphamide, T–docetaxel, H–trastuzumab, L–lapatinib.

**Figure 2 cancers-11-01186-f002:**
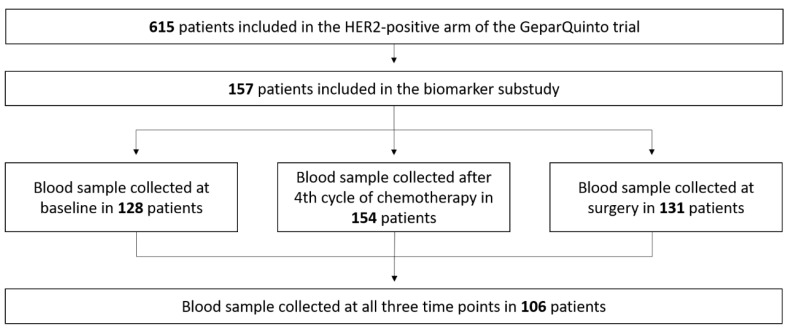
Flow diagram showing the number of patients with a blood sample collected at different time points.

**Figure 3 cancers-11-01186-f003:**
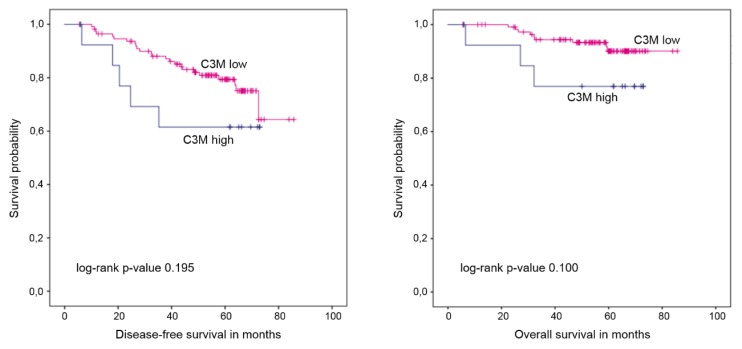
Kaplan–Meier plots of disease-free and overall survival stratified by C3M levels at baseline.

**Figure 4 cancers-11-01186-f004:**
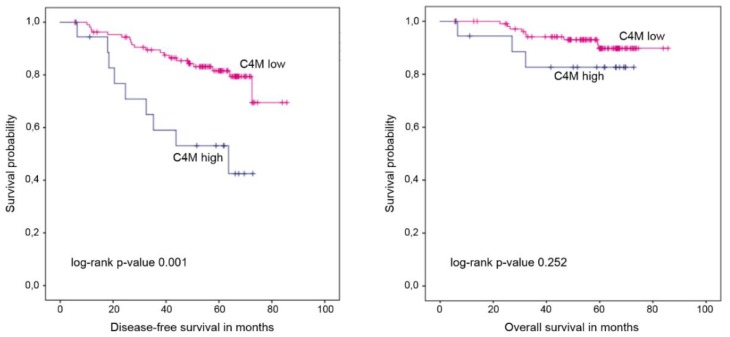
Kaplan–Meier plots of disease-free and overall survival stratified by C4M levels at baseline.

**Figure 5 cancers-11-01186-f005:**
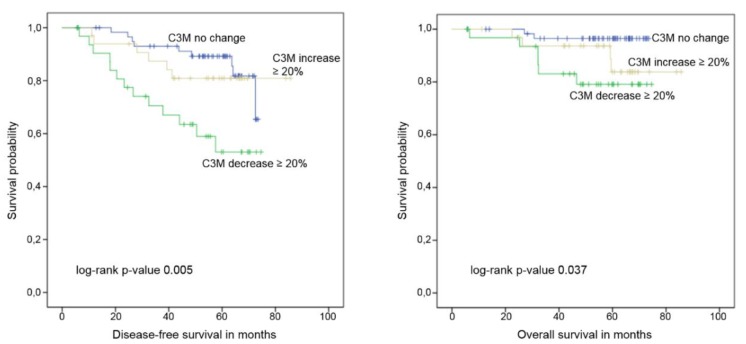
Kaplan–Meier plots of disease-free and overall survival stratified by changes in C3M levels between baseline and after four cycles of therapy (‘no change’ is defined as an increase or decrease <20%).

**Table 1 cancers-11-01186-t001:** Correlation between collagen degradation marker levels in the serum at baseline and clinical-pathological, treatment and outcome variables.

Parameter	Total	C3M			C4M		
		High C3M n (%)	Low C3M n (%)	*p*-Value ^1^	High C4M n (%)	Low C4M n (%)	*p*-Value ^1^
Overall	128	15 (11.7%)	113 (88.3%)		19 (14.8%)	109 (85.2%)	
Age							
>50 years	66	8 (12.1%)	58 (87.9%)	1.000	10 (15.2%)	56 (84.8%)	1.000
≤50 years	62	7 (11.3%)	55 (88.7%)		9 (14.5%)	53 (85.5%)	
cT stage							
cT1/2	75	9 (12.0%)	66 (88.0%)	1.000	10 (13.3%)	65 (86.7%)	0.616
cT3/4	52	6 (11.5%)	46 (88.5%)		9 (17.3%)	43 (82.7%)	
cN stage							
cN0	51	6 (11.8%)	45 (88.2%)		6 (11.8%)	45 (88.2%)	
cN+	75	9 (12.0%)	66 (88.0%)	1.000	13 (17.3%)	62 (82.7%)	0.455
Estrogen receptor status							
Positive	64	5 (7.8%)	59 (92.2%)	0.271	8 (12.5%)	56 (87.5%)	0.620
Negative	64	10 (15.6%)	54 (84.4%)		11 (17.2%)	53 (82.8%)	
Progesterone receptor status							
Positive	53	3 (5.7%)	50 (94.3%)		7 (13.2%)	46 (86.8%)	
Negative	75	12 (16.0%)	63 (84.0%)	0.096	12 (16.0%)	63 (84.0%)	0.802
Grading							
G1–2	65	5 (7.7%)	60 (92.3%)	0.177	7 (10.8%)	58 (89.2%)	0.220
G3	63	10 (15.9%)	53 (84.1%)		12 (19.0%)	51 (81.0%)	
Anti-HER2 therapy							
Lapatinib	63	6 (9.5%)	57 (90.5%)	0.585	9 (14.3%)	54 (85.7%)	1.000
Trastuzumab	65	9 (13.8%)	56 (86.2%)		10 (15.4%)	55 (84.6%)	
pCR							
Yes	39	10 (25.6%)	29 (74.4%)	**0.002**	10 (25.6%)	29 (74.4%)	**0.031**
No	89	5 (5.6%)	84 (94.4%)		9 (10.1%)	80 (89.9%)	

^1^ Fisher’s Exact Test (2-sided). Abbreviations: pCR–pathological complete response.

**Table 2 cancers-11-01186-t002:** Levels of C3M and C4M in the serum at different time points.

Time Point of Blood Sampling	Value	C3M (ng/mL)			C4M (ng/mL)		
		Total	Trastuzumab Arm	Lapatinib Arm	Total	TRASTUZUMAB ARM	Lapatinib Arm
Baseline	Median	6.364	6.316	6.636	31.940	32.180	31.724
	Mean	6.597	6.588	6.606	33.148	34.058	32.209
	Range	1.640–17.236	3.128–17.236	1.640–12.264	15.940–71.684	15.940–71.684	17.816–69.192
After 4 cycles of neoadjuvant therapy	Median	6.100	6.036	6.172	33.664	33.856	33.304
	Mean	6.295	6.148	6.435	34.500	34.246	34.740
	Range	3.128–12.012	3.128–11.388	3.624–12.012	14.020–87.176	15.680–67.700	14.020–87.176
At time of surgery	Median	5.352	5.352	5.290	30.604	30.742	29.814
	Mean	5.747	5.826	5.675	32.760	33.632	31.952
	Range	1.724–14.316	1.724–13.132	2.496–14.316	14.784–78.996	18.416–70.060	14.784–78.996

**Table 3 cancers-11-01186-t003:** Changes in serum levels of C3M and C4M between baseline and surgery according to treatment arm.

Changes in Serum Levels	C3M n (%)			C4M n (%)		
	Total	Trastuzumab	Lapatinib	Total	Trastuzumab	Lapatinib
Increase ≥20%	24 (22.6%)	12 (23.1%)	12 (22.2%)	25 (23.6%)	12 (22.6%)	13 (24.5%)
No change ^1^	43 (40.6%)	20 (38.5%)	23 (42.6%)	58 (54.7%)	29 (54.7%)	29 (54.7%)
Decrease ≥20%	39 (36.8%)	20 (38.5%)	19 (35.2%)	23 (21.7%)	12 (22.6%)	11 (20.8%)

^1^ defined as an increase <20% or decrease <20%.

**Table 4 cancers-11-01186-t004:** Survival times in correlation with C3M and C4M levels at baseline.

Endpoint		C3M High vs. Low	C4M High vs. Low
Disease-free survival	Median	Not reached vs. not reached	63.7 months vs. not reached
	Mean	53.0 (95%-CI 38.9–67.1) vs. 72.2 (95%-CI 67.2–77.2) months	49.5 (95%-CI 37.9–61.0) vs. 74.0 (95%-CI 69.2–8.8) months
	Log rank *p*-value	0.195	0.001
	5-year DFS	61.5% vs. 79.4%	53.1% vs. 81.6%
	HR (95% CI), *p*-value	1.88 (0.71–4.95), *p* = 0.202	3.39 (1.54–7.45), *p* = 0.002
Overall survival	Median	Not reached vs. not reached	Not reached vs. not reached
	Mean	61.2 (95%-CI 49.2–73.3) vs. 81.2 (95%-CI 78.4–84.0) months	64.0 (95%-CI 54.8–73.3) vs. 81.0 (95%-CI 78.1–84.0) months
	Log rank *p*-value	0.100	0.252
	5-year OS	76.9% vs. 90.1%	82.6% vs. 89.8%
	HR (95% CI), *p*-value	2.86 (0.77–10.57), *p* = 0.116	2.11 (0.57–7.80), *p* = 0.263

Abbreviations: DFS–disease-free survival; HR–hazard ratio; OS–overall survival.

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
