# Peer review of "Clinical Relevance of Collagen Protein Degradation Markers C3M and C4M in the Serum of Breast Cancer Patients Treated with Neoadjuvant Therapy in the GeparQuinto Trial"

_cancers, 2019, doi:10.3390/cancers11081186_

Round 1

Reviewer 1 Report

In this study, the authors present a correlation between serum levels of C3M and C4M (as collagen degradation markers) and survival, pCR rates, etc. in HER2+ patients who received neoadjuvant therapy from a clinical trial.

The results are interesting and can be published after the following minor revisions:

Line 87: sentence should be clarified. It looks like “Out of 615 patients, 157 patients were selected?”

Table 1: What is the reason for the discrepancies in the number of patients between the 3 arms and then the final number? Line 232 touches on the small sample size but an explanation would be good.

The Introduction mentions C1M, C3M, and C4M as markers. So I am curious as to why only C3M and C4M were assayed? This can be pointed out in the results or discussion.

It will help if Fig 5 is the first or second figure, when it is mentioned in the Results text. It will clarify the experimental design early on in the reading. Also, the full forms of EC, H, T, etc should be included in the captions. Is it unclear what H is. If H is docetaxel, wasn’t it given to both the groups?

Line 94: It will help to state the actual levels in healthy females. These are mentioned on Lines 278-279 but a direct comparison in Results is convenient.

Can the authors comment on any connection between HER2 expression and collagen degradation and MMP in the Discussion?

Fig 2,3,4. What is the n for each of the groups? Or the percentile?

The fact that baseline levels of these markers are higher than in healthy patients is not surprising. But it is interesting that higher levels correlate with higher rate of pCR. Can the authors comment on how the levels of the specific MMPs may change with the specific neoadjuvant therapeutic agents (targeted at HER2+ tumors) in this study?

Minor proofreads

Author Response

We would like to thank the Reviewers for their constructive and insightful comments. Attached please find our point-by-point responses to the Reviewers’ suggestions and a revised manuscript according to the points addressed.

Reviewer 1: In this study, the authors present a correlation between serum levels of C3M and C4M (as collagen degradation markers) and survival, pCR rates, etc. in HER2+ patients who received neoadjuvant therapy from a clinical trial. The results are interesting and can be published after the following minor revisions:

Line 87: sentence should be clarified. It looks like “Out of 615 patients, 157 patients were selected?”

We changed the sentence to make it clearer.

Table 1: What is the reason for the discrepancies in the number of patients between the 3 arms and then the final number? Line 232 touches on the small sample size but an explanation would be good.

The GeparQuinto trial was one of the largest neoadjuvant trials conducted worldwide (> 2600 pts). Of these, 615 were HER2-positive. These patients were offered to take part in the C3M/C4M substudy. However, there were several other subprojects available so not all patients in the HER2-positive arm could be recruited. We expanded the explanation in the Discussion accordingly.

The Introduction mentions C1M, C3M, and C4M as markers. So I am curious as to why only C3M and C4M were assayed? This can be pointed out in the results or discussion.

We included a short explanation in the discussion. Kits for the two markers (deemed most promising in terms of therapy response prediction) were provided at no cost for this substudy by the manufacturer.

It will help if Fig 5 is the first or second figure, when it is mentioned in the Results text. It will clarify the experimental design early on in the reading. Also, the full forms of EC, H, T, etc should be included in the captions. Is it unclear what H is. If H is docetaxel, wasn’t it given to both the groups?

We changed the position of Fig 5 and explained the abbreviations. In the diagram, H is trastuzumab / herceptin.

Line 94: It will help to state the actual levels in healthy females. These are mentioned on Lines 278-279 but a direct comparison in Results is convenient.

Done.

Can the authors comment on any connection between HER2 expression and collagen degradation and MMP in the Discussion?

Done.

Fig 2,3,4. What is the n for each of the groups? Or the percentile?

All patients with known C3M/C4M status were included in this analysis.

The fact that baseline levels of these markers are higher than in healthy patients is not surprising. But it is interesting that higher levels correlate with higher rate of pCR. Can the authors comment on how the levels of the specific MMPs may change with the specific neoadjuvant therapeutic agents (targeted at HER2+ tumors) in this study?

We expanded this part of the discussion.

Reviewer 2 Report

This clinical trial paper reported collagen protein degradation markers C3M and C4M as bio-markers for breast cancer patients who were treated with chemotherapy. Although the sample size is not large, the statistical difference still could be observed.

In this study, overall survival and disease-free survival were evaluated with C3M and C4M levels in patients' serum. However, the aggressive feature of different subtype tumor and the tumor sizes of patients have not been investigated in this study as well as. If it is possible, it is better to present the tumor size or relevant parameters at the blood collection time points. This will provide more information on whether levels of C3M and C4M is correlated with tumor feature during or after chemotherapy.

Author Response

We would like to thank the Reviewers for their constructive and insightful comments. Attached please find our point-by-point responses to the Reviewers’ suggestions and a revised manuscript according to the points addressed.

This clinical trial paper reported collagen protein degradation markers C3M and C4M as bio-markers for breast cancer patients who were treated with chemotherapy. Although the sample size is not large, the statistical difference still could be observed. In this study, overall survival and disease-free survival were evaluated with C3M and C4M levels in patients' serum. However, the aggressive feature of different subtype tumor and the tumor sizes of patients have not been investigated in this study as well as. If it is possible, it is better to present the tumor size or relevant parameters at the blood collection time points. This will provide more information on whether levels of C3M and C4M is correlated with tumor feature during or after chemotherapy.

This is a very good point. However, data on the tumor size during treatment, measured e.g. by ultrasound, was not collected in this trial so we cannot comment on any possible association between response to treatment (observed upon imaging) and levels of serum markers.

Reviewer 3 Report

The authors present a study about the correlation between the collagen degradation markers and the response to neoadjuvant therapy, and its role in breast cancer.  The study is interesting and relevant but can be improved. I have the following recommendations:

1 - A rigorous statistical analysis must be carried out before publication:

1.2- The authors must explain in more detail the characteristics of the control group used. Please include this in "Materials and Methods" section;

1.3 - Also, the information in the table 1 must be completed including a statistical analysis using the Tukey's HSD (honestly significant difference) test and discuss the differences with Fisher’s Exact Test in discussion section;

2 - The information contained in table 1, could be clearer if the authors present it in a bar graph, in which the statistical differences are indicated with superscript-letters;

3- If a Tukey's test is included, it will be important to include for readers the importance and how both tests (Tukey's test and fisher’s test) complement each other. So I recommend the following reading to include some ideas in the "3.2-Statistical analysis" section: doi: 10.4097/kja.d.18.00242

Author Response

We would like to thank the Reviewers for their constructive and insightful comments. Attached please find our point-by-point responses to the Reviewers’ suggestions and a revised manuscript according to the points addressed.

The authors present a study about the correlation between the collagen degradation markers and the response to neoadjuvant therapy, and its role in breast cancer.  The study is interesting and relevant but can be improved. I have the following recommendations:

1 - A rigorous statistical analysis must be carried out before publication:

The statistical analysis was performed by full-time statisticians employed by the German Breast Group, one of the largest study groups worldwide. This particular study, the GeparQuinto trial, is among the largest neoadjuvant breast cancer trials ever conducted (2600 patients in total).

Another part of the translational substudy within this trial has been published in the Brit J Cancer 2012 (Witzel et al., Predictive value of HER2 serum levels in patients treated with lapatinib or trastuzumab – a translational project in the neoadjuvant GeparQuinto trial). The statistical analysis and the tests conducted were the same as in the present study on C3M/C4M.

1.2- The authors must explain in more detail the characteristics of the control group used. Please include this in "Materials and Methods" section;

We included Supplementary Table 3 with these data.

1.3 - Also, the information in the table 1 must be completed including a statistical analysis using the Tukey's HSD (honestly significant difference) test and discuss the differences with Fisher’s Exact Test in discussion section

2 - The information contained in table 1, could be clearer if the authors present it in a bar graph, in which the statistical differences are indicated with superscript-letters;

3- If a Tukey's test is included, it will be important to include for readers the importance and how both tests (Tukey's test and fisher’s test) complement each other. So I recommend the following reading to include some ideas in the "3.2-Statistical analysis" section: doi: 10.4097/kja.d.18.00242

See above.

Reviewer 4 Report

The article is well written and describe clearly objectives and results. Also, Authors addressed the study limitations.

There are minor revisions to be fixed before consider the manuscript worthy for publication

Line 88. Authors stated that there are no differences among the two arms. Could you show descriptive characteristics of the patients enrolled in the arms? Maybe as supplementary table. Line 255 could you provide the approval ID? Table 2. Could you provide data related to the collagen degradation markers after the neoadjuvant therapy, also stratified by treatment? Could you provide the protocol registration number? Could you provide some considerations on future perspectives? Which are the implications of your results in clinical practice?

Author Response

We would like to thank the Reviewers for their constructive and insightful comments. Attached please find our point-by-point responses to the Reviewers’ suggestions and a revised manuscript according to the points addressed.

The article is well written and describe clearly objectives and results. Also, Authors addressed the study limitations. There are minor revisions to be fixed before consider the manuscript worthy for publication. Line 88. Authors stated that there are no differences among the two arms. Could you show descriptive characteristics of the patients enrolled in the arms? Maybe as supplementary table.

This is a good point. We included a supplementary table.

Line 255 could you provide the approval ID?

Done.

Table 2. Could you provide data related to the collagen degradation markers after the neoadjuvant therapy, also stratified by treatment?

We expanded Table 2 accordingly.

Could you provide the protocol registration number?

We included the EudraCT number and the exact approval number as well.

Could you provide some considerations on future perspectives? Which are the implications of your results in clinical practice?

We added the section of Conclusions and commented on some future directions.